# Increased Scabies Incidence at the Beginning of the 21st Century: What Do Reports from Europe and the World Show?

**DOI:** 10.3390/life12101598

**Published:** 2022-10-13

**Authors:** Marija Delaš Aždajić, Iva Bešlić, Ana Gašić, Nikola Ferara, Lovre Pedić, Liborija Lugović-Mihić

**Affiliations:** 1Department of Dermatovenereology, University Hospital Centre Sestre Milosrdnice, 10000 Zagreb, Croatia; 2Department of Dermatovenereology, General Hospital Šibenik, 22000 Šibenik, Croatia; 3Department of Dermatovenereology, General Hospital Zadar, 23000 Zadar, Croatia; 4School of Dental Medicine, University of Zagreb, 10000 Zagreb, Croatia

**Keywords:** scabies, epidemiology, epidemiology trends, risk factors, diagnostics, reports

## Abstract

Reports from various countries have described increasing numbers of scabies cases, especially in the past two decades. The epidemiological data for various world regions showed prevalence estimates ranging from 0.2% to 71%, with the highest prevalence in the Pacific region and Latin America. Therefore, geographically, scabies occurs more commonly in the developing world, tropical climates, and in areas with a lack of access to water. According to results from specific regions of the world, the greatest burdens from scabies were recorded for East Asia, Southeast Asia, Oceania, tropical Latin America, and South Asia. Among countries with the highest rates, the top 10 were Indonesia, China, Timor-Leste, Vanuatu, Fiji, Cambodia, Laos, Myanmar, Vietnam, and the Seychelles. From Europe, available data shows an increasing trend in scabies infestation, particularly evident among populations with associated contributing factors, such as those who travel frequently, refugees, asylum seekers, those who regularly lack drinking water and appropriate hygiene and are of a younger age, etc. This increase in observed cases in the last 10–20 years has been evidenced by research conducted in Germany, France, Norway, and Croatia, among other countries. In addition, increased scabies transmission was also recorded during the COVID-19 pandemic and may have been the result of increased sexual intercourse during that time. Despite all the available treatment options, scabies commonly goes unrecognized and is therefore not treated accordingly. This trend calls for a prompt and synergistic reaction from all healthcare professionals, governmental institutions, and non-governmental organizations, especially in settings where population migration is common and where living standards are low. Furthermore, the proper education of whole populations and accessible healthcare are cornerstones of outbreak prevention. Accurate national data and proper disease reporting should be a goal for every country worldwide when developing strategic plans for preventing and controlling the community spread of scabies.

## 1. Introduction

For centuries, scabies has primarily been thought of as a disease that affects those living in squalor and poverty; however, reports in recent times show that it has become more common in the general population. Generally, scabies has been thought to occur only sporadically, but in the last two decades, there has been an increase in published papers indicating that disease is occurring more frequently [1,2,3,4,5,6,7,8,9].

Reports from various countries have described an increase in the incidence of scabies (Table 1) [8,10,11,12,13,14,15,16,17,18,19]. From Europe, available data shows an increasing trend in scabies infestations, particularly evident among populations with associated contributing factors, such as those who travel frequently, refugees, asylum seekers, those who regularly lack drinking water and appropriate hygiene, those who are of a younger age, etc. This increase in observed cases in the last 10–20 years has been evidenced by research conducted in Germany, France, Norway, and Croatia, among other countries (Table 2) [1,2,5,6,7,20,21,22,23,24,25,26,27,28,29,30,31]. In addition, in recent years, there has been greater public discussion around scabies, and in 2017, the World Health Organization (WHO) listed scabies as a Neglected Tropical Disease (NTD) [32]. Therefore, the current scientific literature consists of an increasing number of published papers concerning the frequency of scabies in the world and Europe. At the same time, physicians and dermatovenereologists have been reporting increased cases in their daily practices. For this reason, we wanted to present the current data available on the frequency of and risk factors for scabies by looking at studies (available in the PubMed database) published during the period between 2000 and 2022 (i.e., at the beginning of the 21st century).

## 2. Clinical Features and Management of Scabies

Scabies is a common, contagious ectoparasite infection caused by the mite *Sarcoptes scabiei var. hominis* [33,34,35,36]. The disease may present with typical signs of itching (predominantly) and lesions at certain predilection sites (the sides and webs of the fingers, wrists, axillae, areolae, umbilical area, genitalia, etc.) (Figure 1). However, scabies may also have an atypical presentation or be similar to other diseases [37]. A rarer type, crusted scabies, is associated with a heavy mite burden and usually manifests with thick scales, crusts, and fissures. This type of scabies was formerly known as Norwegian scabies and primarily occurs in older adults or immunocompromised individuals [38]. The transmission of scabies usually happens through close, person-to-person skin contact, and a true scabies infestation is not transmitted from animals to humans (animal infestation is caused by a slightly different mite, *Sarcoptic Mange*) [37,39].

The clinical manifestation of a primary scabies infestation can take from 2 to 6 weeks to appear, whereas reinfestation can trigger a swift allergic reaction [40]. The most common scabies manifestations include rashes, followed by multiple papules and vesicles or even urticaria on specific sites [33,41,42]. Symptoms appear due to an allergic reaction to the mites [33]. Sometimes, though rarely, scabies may be complicated by secondary *Staphylococcal* or *Streptococcal* infections, including impetigo, ecthyma, paronychia, and furunculosis. *Streptococcal* infections may further lead to *poststreptococcal* glomerulonephritis or other complications, such as nephritis, acute rheumatic fever, or fatal invasive sepsis [43,44].

Since scabies is a contagious skin disease which spreads by direct contact, the most important aspect of its management is timely recognition and reporting, which is currently often inadequately performed [33,40]. It is crucial to include/consider scabies as one of many differential diagnoses in cases when an itch is associated with eczematous lesions and/or findings of skin burrows or comma-like papules at scabies-typical characteristic localizations. Furthermore, in patients presenting with pruritus (especially a nocturnal itch), scabies should always be considered [45].

The Consensus Criteria for the Diagnosis of Scabies published in 2020 by The 2020 International Alliance for the Control of Scabies comprises three degrees of diagnostic certainty: confirmed scabies, clinical scabies, and suspected scabies [39]. While a suspicion of scabies infestation is based upon patient history and a physical examination, confirmation by the parasitological examination of mites, eggs, or faeces (skin preparation) is crucial for further therapy and outcomes [46]. Some patients can have a false negative finding through skin scrapings, leading to problems in practice when patients are treated ineffectively [33,47]. However, clinicians commonly set a diagnosis based only on the clinical picture, specific localization, and itch [9,21]. In addition, dermoscopic findings may be a useful diagnostic tool in daily practice (Figure 2). One multicentre study conducted among several European countries reported that clinicians predominantly confirmed scabies infestation based on clinical presentation [31]. Additionally, a study conducted in the United Kingdom reported that scabies outbreaks in care homes were always diagnosed clinically and by general practitioners or home staff (not by dermatologists) [30]. According to a French study, general practitioners mainly relied on only the typical localization of pruritus to make a scabies diagnosis, while diagnostic tests were used by just 6% of practitioners (at least one or more scabies cases was reported by 89% of general practitioners) [21]. So, most physicians do not perform essential diagnostic procedures or do not have the available essential diagnostic tools and, consequently, anti-scabies therapy is often given only on the basis of a clinical appearance. This can be problematic in practice, especially in modern/Western countries because patients may be disinclined to accept treatment without diagnostic evidence of scabies, especially those who do not belong to a risk group. However, when a parasitological examination is not easily accessible, the clinician’s experience and a patient’s psychological and physical profile and quality of life may help in the early recognition of scabies in rural areas [48].

According to recommendations and available data, treatment for classic scabies includes a 5% permethrin cream or a 25% benzyl benzoate lotion, or sometimes alternative treatments (a 0.5% malathion aqueous lotion, 1% ivermectin lotion, and 6–33% sulphur cream, ointment, or lotion) [49]. Oral ivermectin is also effective for treating scabies [50]. For crusted scabies, therapy includes a combination of topical scabicide and oral ivermectin [50,51]. For the mass treatment of large populations where scabies is endemic, single doses of ivermectin can be administered [51]. However, resistance to anti-scabies therapy has often been recorded in practice [49,50]. Persistent symptoms caused by a scabies infestation raise the need to reconsider the diagnosis and treatment options. They can be the result of several causes, including a misdiagnosis, insufficient treatment, an inappropriate drug prescription, poor compliance or noncompliance with treatment, post-scabetic reactions to the mites or their products, reinfection, delusions of parasitosis, or a drug-resistant case of scabies. There has been an increasing trend in drug-resistant scabies—which is the result of long-term use or overdosing of scabicides, resulting in prolonged treatment procedures, repeated visits to healthcare providers, high healthcare costs, and social stigmatization for an ever-greater number of patients [49,50]. The management of scabies includes not only the need to treat the patient successfully but to control the transmission of the disease. Scabies infestation can be prevented by avoiding direct skin-to-skin contact with an infested person and their personal items. The mean survival time of the mite outside of the host is between 48 and 72 h; therefore, items used by infested person within this period should either be placed in a plastic bag for at least 72 h or should be machine washed in hot water (at least 60 °C) and machine dried or dry-cleaned. Simultaneous scabies treatment is recommended both for patients and their close contacts, particularly those who have had prolonged skin-to-skin contact with an infested person, regardless of symptoms.

## 3. General Epidemiological Aspects of Scabies

Epidemiologically, it is estimated that the worldwide prevalence of scabies is 200–300 million people, with wide variations among specific geographic regions [52]. The epidemiological data (based on the systematic review of population-based studies) for various world regions, except North America, showed prevalence estimates ranging from 0.2% to 71%, with the highest prevalence in the Pacific region and Latin America. Therefore, geographically, scabies occurs more commonly in the developing world, tropical climates, and in areas with a lack of access to water [53].

By age and gender, it is generally equally common in both sexes and among people of different ages [4,54]. However, by age, data shows that scabies more commonly affects children and young adults [4,5,22,24,54,55,56]. Concerning gender, variations have also been described—some parts of the world note a higher prevalence among women and others among men [14,17,43,57,58].

Epidemiologically, scabies is considered a disease of those who live in poor socioeconomic conditions, yet the disease affects individuals of any socioeconomic status [33,40]. Still, the risk of transmission increases in crowded living conditions, resource-limited regions, child-care facilities, group homes, and institutional settings (e.g., long-term care facilities, prisons, etc.) [33]. Furthermore, recent data have confirmed a correlation between scabies incidence and population movements, meaning that population movements lead to a higher incidence in the general population [4,59,60,61,62]. According to study results, among refugee and asylum seeker populations, scabies is one of the three most frequently reported infectious diseases [23]. According to results of a recent study that included statistical analyses, among the determinant factors of scabies outbreaks were travel in the last six weeks to an area experiencing a scabies epidemic, the presence of person with itching in the family/household, sleeping with a scabies patient, and the infrequent use of a detergent when showering [63]. According to another study, scabies was among the most common skin diseases diagnosed in travellers returning from the tropics, along with infectious cellulitis, pruritis of unknown origin, and others [64].

## 4. Trends in Scabies Incidence/Prevalence Recorded during the Last Two Decades

As of 2015, scabies affected 2.8% of the population worldwide, i.e., 200–300 million people [34,52,65]. During the 1960s, and peaking around 1980, scabies became more frequent in North America and Europe. It has declined somewhat but remains a common problem. Generally, scabies is underreported in most of countries, and its management is often inadequate [40]. Although there are few global epidemiological studies, some research analyses and systematic reviews (population-level and country-level) support/indicate certain trends [10,66]. Therefore, due to the increasing trend of scabies prevalence and lack of clinical recognition, the WHO has identified scabies as a neglected tropical disease in 2017 and is encouraging countries to stop this trend by 2030 as part of its Sustainable Development Goals [66].

Karimkhani, C. et al. estimated the global burden of scabies through the analysis of data from 195 countries between 1990 to 2015 [10]. Their cross-sectional analysis combined prevalence estimates with a disability weight, measuring disfigurement, itch, and concomitant pain in order to determine years lived with disability (YLDs); where YLDs were equivalent to disability-adjusted life-years (DALYs). According to results from specific regions of the world, the greatest burdens (DALYs) from scabies were recorded for East Asia (age-standardised DALYs 136·32), Southeast Asia (134·57), Oceania (120·34), tropical Latin America (99·94), and South Asia (69·41). Among countries with the highest rates (DALY burdens per 100,000 people), the top 10 were Indonesia (153·86), China (138·25), Timor-Leste (136·6), Vanuatu (131·59), Fiji (130·91), Cambodia (126·93), Laos (124·96), Myanmar (124·46), Vietnam (123·30), and the Seychelles (122·99) [10]. So, the burden of scabies over the human lifespan varies according to region and a wide analysis revealed that scabies burden is greatest in tropical regions, such as the above-mentioned parts of Asia, Oceania, and Latin America, and varies by age (especially common in children, adolescents, and the elderly). Specifically, the greatest scabies burdens by age were recorded for East and Southeast Asia, with highest burden (DALY) for ages 1–4 years, followed by a gradually decreasing rate for ages 5–24 years and into adulthood, followed by a slight increase after age 70. This pattern is much less common in North America and western Europe, as these are low-burden regions, where its prevalence is more evenly distributed across all age groups, including elderly people, though scabies outbreaks can be common in care homes.

Although scabies prevalence has never been particularly high in Europe, a trend of increasing cases has been reported. German data show that scabies case numbers have been rising since 2009 and especially since 2014. Among outpatients (during 2010–2015), there was an increase of 52.8% to around 128,000 treatment cases; in Germany, more than 11,000 inpatient scabies cases were recorded annually (as the main diagnosis) [7]. In addition, during the period between 2010 and 2016, there was an increase of about 306%. While outpatients were predominately treated by dermatologists and general practitioners, inpatients were treated in dermatology, paediatrics, and internal medicine departments. According to data from the literature, the prevalence of scabies infestation is higher in immigrants and refugees coming from the Middle East and Africa than in the general German and Belgian populations [1,67]. A German study conducted by Augustin et al. [7] confirmed similar findings, and the same trend was recognized by another German study conducted by Reichert F et al., who reported a nine-fold increase in Germany in the period between 2009 and 2018 [22]. Similarly, data from a Spanish study confirms the German findings: Redondo-Bravo et al. reported increased scabies admissions in the period between 2014 and 2017, which they explained by a worsening in living conditions due to impacts of the economic crisis and the presence of migrant groups [28]. Thus, in Spain, they recorded a decreasing rate of patient admissions due to scabies during 1997–2014, followed by an increasing trend from 2014 to 2017. Additionally, wide geographical differences were recorded, which were dependent on the database explored. Similar results were found in Norway, where authors reported an increase in scabies diagnoses since 2013 [2]. In addition, data from the Netherlands confirms that the scabies incidence in that country had increased by more than threefold for the period from 2011–2020 [5].

Factors that contribute to an increase in scabies occurrence are travel and refugee status. Outbreaks have been noted in asylum centres across Western Europe, which is unsurprising given that direct, prolonged skin-to-skin contact, in this case due to overcrowding, contributes to scabies transmission [40]. Geographical location can also contribute to an increased scabies incidence, especially in the case of mass migration, e.g., along refugee migration routes. Thus, in Croatia, an approximately 6-fold increase in scabies incidence was observed between 2007–2017 during numerous refugee/migrant movements from the Middle East to western Europe, where counties near the south-eastern European migrant route were particularly affected [4]. However, this upward trend of scabies in Croatia began in 2008, with the steepest increase during 2014–2017, which could be explained by late disease diagnostics and recognition, the delay of anti-scabies treatment, as well as an increase of migration from neighbouring countries with high scabies incidence rates (Bosnia and Hercegovina, Kosovo, Serbia), and an increase in travel and tourism [4].

Increasing trends have also been recorded in other countries. Reports from Istanbul, Turkey, show tertiary-centre scabies outbreaks in 2018 and 2019 [29], along with a rising trend of scabies infestation [55]. Greece also noted an increasing trend of scabies infestation for the period from 2016–2020 [23]. When analysing epidemiological trends for scabies incidences, English authors noticed a cyclical increase of scabies infestation every 20 years [56,59]. Another important observation was the finding of seasonal variations in scabies occurrences [19,68]. According to available data, scabies infestations are especially common in winter [4,19,26,56]. In addition, scabies outbreaks are commonly noted in nursing homes and extended-care facilities, places where timely diagnoses are especially important so that there is less risk of any accompanying complications [1]. These observations indicate areas of risk for scabies transmission, and preventive measures are needed for these situations and settings.

Concerning data on scabies occurrence during the COVID-19 pandemic, some interesting observations have been recorded. One Italian clinic reported a significant increase in the percentage of patients positive for scabies, especially for people under 18 and over 65 during a governmental “stay-at-home” policy period (March 2020–March 2021) [25]. Similar reports were confirmed in Turkey, Spain, Ireland, and some Italian counties/centres, where investigators observed a rising trend of scabies infestation during COVID-19 lockdown periods [3,24,27,69,70]. This rising trend is likely due to close human-to-human contact during the lockdown policies and the lack of healthcare accessibility [71]. Furthermore, scabies transmission during the COVID-19 pandemic may have been the result of increased sexual intercourse since there was a noted increase in the frequency of several sexually transmitted diseases during that time [69]. French authors, however, noticed a decrease in the sale of local and oral anti-scabies therapies during the same period, which they explained as a result of physical distancing and the reduction of scabies circulation [20].

It should be noted, however, that due to a lack of reports based on the specific national data, and predominantly local reports, it is often not possible to obtain precise data.

## 5. Measures for Improvement of Negative Trends

Scabies is a significant and common health problem, a highly contagious disease which can affect quality of life, work performance, sleep, and cause/influence psychosocial problems in patients and their family members. Therefore, additional measures in primary care and the public health management of scabies could impact overall patient quality of life and reduce this infectious disease in whole communities [72].

Although scabies represents an increasing public health problem, until recently, there were no agreed-upon international diagnostic guidelines [73]. Authors from the United Kingdom performed an analysis of 20 different guidelines and noted the heterogeneous nature of these guidelines, especially concerning prophylactic measures and anti-scabies treatment, infection control measures, and mass treatment [74]. Thus, a very useful epidemiological measure would be to publish globally agreed-upon guidelines on measures for scabies prevention and management. Due to an increase in the scabies rate/incidence associated with large migrations, German guidelines published in 2016 contain lists for additional healthcare measures, including the possible treatment of healthy persons/persons who come into contact with infected persons, clothes, linens, or other possibly infected articles [75]. Special attention must also be given to the treatment of infected persons’ sexual partners with a look-back period of 2 months, including screening for other sexually transmitted infections. Patients and their close contacts should avoid sexual contact until anti-scabies therapy completion and should conduct specific personal hygiene measures, especially when in crowded spaces. Finally, medical records should be created and kept even for suspected/unconfirmed cases [37,51]. The latest European guidelines for the management of scabies published in 2017 relies on data from a comprehensive literature review of previous guidelines [51]. In 2020, the IACS Criteria for the Diagnosis of Scabies were established and may provide greater consistency and international standardization for the diagnosis of scabies [39].

Based on outbreak data, the late recognition of scabies by physicians has been a significant problem, e.g., some medical doctors had not even seen scabies cases before and consequently did not take into consideration scabies as a possible diagnosis. Additionally, there are many examples of non-compliance with anti-scabies treatment in practice. Epidemiological services have an important role to play in working out the scope of contacts, stressing the necessity of conducting and overseeing therapy. Therefore, for early recognition and the proper and timely management of patients with scabies, there is a necessity for collaboration between dermatologists, physicians, and epidemiologists.

## 6. Conclusions

Scabies is an easily treatable communicable disease but requires timely diagnostics and treatment to prevent community spread. Unfortunately, the prevalence of this parasitic skin infestation is continuously high and is considered a neglected tropical disease that today affects younger age groups in particular. Recent literature data confirms that many countries have reported increasing numbers of scabies cases, especially in the past two decades. Despite all the available treatment options, the disease commonly goes unrecognized and is therefore not treated accordingly. Continuously growing numbers of patients affected with scabies calls for a prompt and synergistic reaction from all healthcare professionals, governmental institutions, and non-governmental organizations, especially in settings where population migration is common and where living standards are low. Furthermore, the proper education of whole populations and accessible healthcare are cornerstones of outbreak prevention. Finally, accurate national data and proper disease reporting should be a goal for every country worldwide when developing strategic plans for preventing and controlling the community spread of scabies.

## Figures and Tables

**Figure 1 life-12-01598-f001:**
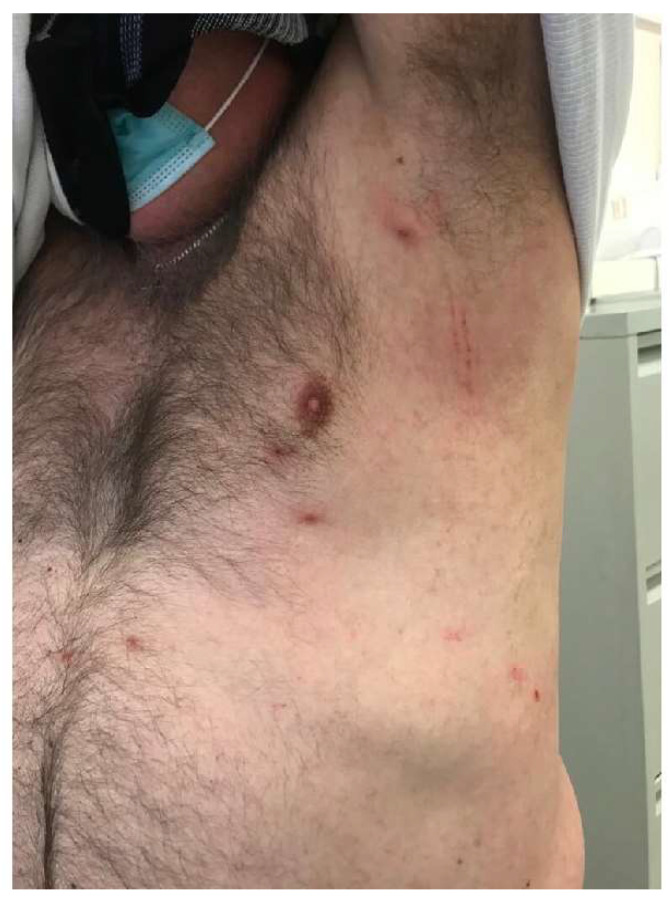
Clinical manifestations of scabies: small erythematous papules accompanied by an intense itching sensation confirmed by linear scratch marks and excoriations scattered throughout the axillar and thoracic region.

**Figure 2 life-12-01598-f002:**
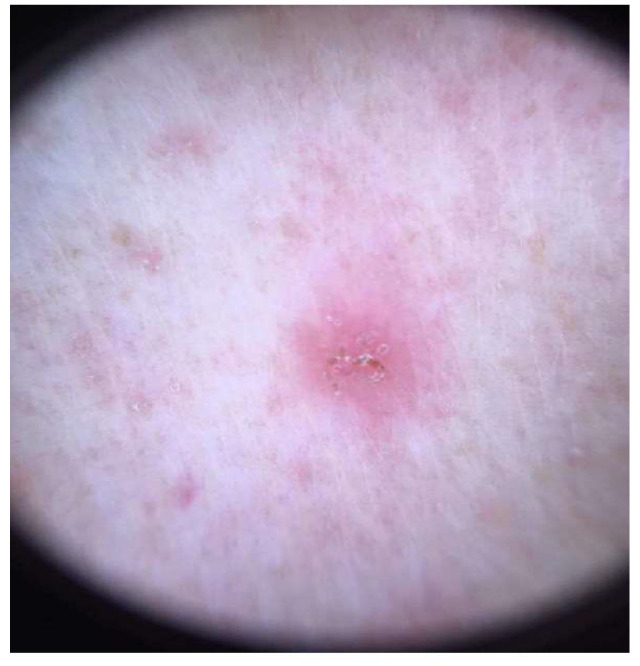
Dermoscopic finding of scabies infestation by the mite *Sarcoptes scabiei*: a triangular dense head structure with an accompanying S-shaped burrow (relatively translucent scabies body) indicate the presence of mites.

**Table 1 life-12-01598-t001:** Selected reports and findings on scabies occurrence in various countries around the world (except Europe) published during 2000–2022.

COUNTRY	AUTHORS	ANALYSED DATA/SOURCE	RESULTS
ARGENTINA	Dei-Cas et al. [8]	A prospective study that included all children aged 15 years or younger who attended the Pediatric Emergency Department of Presidente Peron Hospital (Avellaneda, Buenos Aires, Argentina) between 1 January and 31 December 2016	Scabies was the most frequent parasitic skin infection and fourth most common infectious skin disorder.
CENTRAL AFRICA	Kobangué et al. [11]	Scabies frequency in Bangui assessed using records and clinical confirmation at the dermatology department of Bangui (January 2006–December 2010) (a cross-sectional study)	Average hospital scabies prevalence was 5.88%, highest for those 0–9 years of age (33%). By school age group: preschool age and pupils/students, respectively, 25.5% and 26.3%).
ETHIOPIA	Azene et al. [12]	Systematic review and meta-analysis of overall scabies prevalence and associated factors in Ethiopia; due to high heterogeneity across the studies, the random effects meta-analysis model was used to fix the overall prevalence and associated factors of scabies	Overall scabies prevalence was 14.5% and was significantly associated with (large) family size and bed sharing. The highest prevalence, 19.6%, was seen in the Amhara region.
FIJI	Romani et al. [13]	This mass anti-scabies treatment trial began by looking at the occurrence and predictors of scabies in 2051 participants at baseline from six island communities	High scabies burden and prevalence in Fiji (36.4%), highest in children 5–9 years (55.7%). Important scabies-related factors/findings were overcrowding, young age, and typical clinical distribution.
GHANA	Boateng et al. [14]	Diagnosed skin infections in urban (Greater Accra) and rural (Oti region) areas, reported by study health centres in Ghana (from a 6-month period in 2019)	Among skin infections, 11.6% were scabies cases, the third most common diagnosis behind bacterial dermatitis (26.5%) and tinea (19.5%). Males predominated scabies cases (64.4%).
GLOBAL	Karimkhani et al. [10]	Sources of scabies epidemiological data came from an extensive literature search and hospital insurance records. Data was analysed with a Bayesian meta-regression modelling tool. Estimated DALYs for 195 countries were divided into 21 world regions, 20 age groups, and by both sexes between 1990 and 2015	Scabies was responsible for 0.21% of DALYs from all conditions studied by GBD 2015 worldwide.
MALAWI	Galván-Casas et al. [15]	Scabies frequency in rural Malawi; data from a community-based outreach programme (integrated dermatology clinics and tele-dermatology care, including patient visits, screenings, and treatments)	Total/overall scabies cases increased from 2.9% to 39.2% during 2015–2018; the high prevalence confirmed scabies is a major public health issue in parts of Malawi.
SAUDI ARABIA	Ahmed et al. [16]	Scabies rate in Saudi Arabia and associated factors (a multi-centre retrospective study on adults diagnosed with ≥1 episode of scabies during January 2016–September 2018); 468 adult patients participated	High scabies recurrence rate among adults, with these risk factors: male gender, first tertile (January to April), and living in areas of high humidity. Of patients, 46.8% had recurrences.
SOUTH KOREA	Kim et al. [17]	The national annual and seasonal trend of scabies prevalence based on the National Health Insurance Service database	Annual prevalence (per 1000 persons): those <40 years (0.56–0.69); peak at 3.0–4.1 for persons >80 years; women consistently predominated. High prevalence (>6000 cases) in autumn (when temperatures >25 °C at 2 months prior.
SOLOMON ISLANDS	Lake et al. [18]	Prevalence of scabies was analysed in 20 villages in the Western Province (5239 participants); diagnosis was based on the 2020 International Alliance for the Control of Scabies diagnostic criteria	Overall prevalence was 15.0%; considerable variation by village (3.3–42.6%); highest prevalence in children aged <2 years (27%), and significantly higher prevalence in males (16.7%) than females (13.5%).
TAIWAN	Liu et al. [19]	A nationwide population-based study using data from Taiwan’s National Health Insurance Research Database for the period from January 2000 to December 2013 for a randomly selected sample of one million people from the 23 million people in the database in 2000	The total number of patients infested with scabies was 14,883; the mean age was 52.4 ± 21.0 years. A diagnosis was made according to the patient’s history and a physical examination by a licensed physician.

**Table 2 life-12-01598-t002:** Selected reports and findings on scabies occurrence in European countries published during 2000–2022.

COUNTRY	AUTHORS	ANALYSED DATA/SOURCE	RESULTS
CROATIA	Lugović-Mihić et al. [6]	Retrospective medical records and national data from communicable disease reports during 2007–2017	Six-fold increase in scabies diagnoses in Croatia during 2007–2017, especially in children and young adults; the highest incidence was during 2014–2017 in border counties, probably due to migration flows. The capital, Zagreb, saw a three-fold increase during 2014–2017.
FRANCE	Launay et al. [20]	Data from a healthcare sciences company during lockdown (March–December 2020), including dispensing data of prescription and over-the-counter anti-scabies drugs (from 60% of all French retail pharmacies)	The mean reduction in observed vs. expected sales for topical and oral anti-scabies treatments was 14% and 4%, respectively.
	Schmidt-Guerre et al. [21]	Scabies cases diagnosed during January-June 2015 based on a questionnaire given to general practitioners of the Doubs department in France	GPs frequently diagnosed scabies; 89% had diagnosed ≥1 case during the previous 6 months.
GERMANY	Augustin et al. [7]	Multisource analyses of treatment data from a nationwide statutory health insurance company, the Federal Statistical Office, and data from skin screenings	Scabies case numbers have been rising since 2009, especially since 2014. During 2010–2015 there was a 52.8% increase in outpatients; during 2010–2016 the inpatient increase was about 306%.
	Reichert et al. [22]	Analysis of inpatient and outpatient claims data from 2009 to 2018 in Germany	Scabies diagnoses increased 9-fold during 2009-2018; the highest proportion of scabies was seen in patients 15–24 years old.
	Sunderkötter et al. [1]	Review article on scabies frequency in Germany	Increased scabies diagnoses (no exact quantified data on age-specific and regional incidences); possible regional increases due to migration and an increase in STDs among young adults.
GREECE	Louka et al. [23]	Data from the Greek National Public Health Organization surveillance system for June 2016–July 2020 (retrospective study), submitted by staff at health centres for refugees/asylum seekers	Scabies infestation increased over time, followed by several outbreaks; in the refugee/asylum seeker populations, scabies ranked third among most frequent infectious diseases.
IRELAND	Griffin et al. [24]	Data from a dermatological clinic in Galway for the period March 2020–July 2021 compared to previous 4 years	A significant increase in scabies cases during March 2020–July 2021 compared with averages for the same period of the previous 4 years.
ITALY	De Lucia et al. [25]	Scabies admissions to a dermatological clinic in Naples during lockdown (March 2020–March 2021)	A significant increase in scabies cases, especially among patients <18 and >65 years.
NETHERLANDS	van Deursen et al. [5]	Two national data sources on scabies analysed during 2011–2020	Increased scabies incidence by more than 3-fold during 2011–2020, mainly affecting adolescents and (young) adults.
NORWAY	Amato et al. [2]	Data from Norwegian Syndromic Surveillance System about mite infestations compared with anti-scabies treatment sales during 2006–2018	Increased scabies management and incidence rates since 2013, based on increased consultations/sales of anti-scabies treatments.
POLAND	Korycinska et al. [26]	Data reports on scabies from the Polish National Health Fund for the period 2007–2014	Highest scabies rates for the 10–19 age group; seasonality (predominantly during the autumn and winter months).
SPAIN	Martínez-Pallás et al. [27]	Scabies frequency data during March–May 2020 compared to the averages for the same period for the previous five years in Zaragoza	A significant increase in scabies cases during the confinement period.
	Redondo-Bravo et al. [28]	Nationwide retrospective study of four databases (hospital admissions, patients attended at primary healthcare, outbreaks, and occupational diseases) during 1997–2019	An increasing trend in scabies admissions during 2014–2017; among hospital admissions, the elderly predominated, while children and young adults were treated at primary health centres.
TURKEY	Baykal et al. [29]	Retrospective database search on scabies prevalence from two Istanbul tertiary-level dermatology centres (January 2016–December 2019)	2016 and 2017 saw a stable frequency of scabies diagnoses; 2018 and 2019 saw a scabies outbreak that peaked in the fourth quarter of 2019 (in both centres).
UNITED KINGDOM	Hewitt et al. [30]	Data from seven care homes reporting suspected scabies outbreaks in southern England over a 6-month period (November 2012–April 2013)	Scabies attack rates varied (2−50%). Cases were diagnosed clinically by GPs or care staff (none by dermatologists). Most outbreaks were attributable to delayed diagnoses.
VARIOUS COUNTRIES (Greece, France, The Netherlands, Serbia, Belgium, Turkey, Macedonia)	Richardson et al. [31]	A qualitative study, retrospective semi-structured telephone interviews for scabies data during November 2017–February 2018	Clinicians confirmed scabies by clinical presentation; treatment and outbreak management varied highly.

Abbreviations: GPs—general practitioners; STDs—sexually transmitted diseases.

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
