# Peer review of "Increased Scabies Incidence at the Beginning of the 21st Century: What Do Reports from Europe and the World Show?"

_life, 2022, doi:10.3390/life12101598_

Round 1

Reviewer 1 Report

Thank you for the preparation of this Scabies Review and summary of many recent papers regarding the changing epidemiology. This seems to be the most relevant topic of this review due to the increasing prevalence in some parts of the world.

However, table 1 and 2 need a thorough revision. Many information on data and results are inprecise and hard to understand or even incorrect. These are some examples:

1. Table 1 (Central Africa): what do you mean with "an important achievement of the disadvantaged classes"?
2. Table 1 (Ethiopia): what do you mean by "to fix examined factors"?
3. Table 1 (Fiji): What do you mean by "clinical distribution" as a risk factor?
4. Table 1 (Global): "..." in the author column. Why aren´t you reporting the more recent 2015 GBD Study?
5. Table 1 (Taiwan): was "clinical severity" referring to scabies or was clinical severity of other diseases associated with scabies infections?

6. Table 2 title is referring to publications from 2000-2021 but several studies were referenced with 2022.
7. Table 2 (Augustin et al.): you report the case number increase in outpatients for 2010-2015. Then you report a much higher increase for the period 2010-2016. These numbers probably refer to inpatients but you need to state this.
8. Table 2 (Reichert et al.): "patients were generally 15-24 years old" is inprecise. You probably mean that incidence was highest among 15-24 year-olds?
9. Table 2 (Louka et al.): all data is referring only to health centres for refugees/asylum seekers if I am not mistaken. Your report reads like scabies infestations increased in the general populations. Please specify.

What are the selection criteria for  the studies in table 1 ("prominent reports")? It seems to be a mixture of qualitative studies and quantitative studies on frequency, risk factors but also quality of life. I woud recommend to focus on specific topics or report more extensively (For example also studies on frequency in Brazil).

In line 122/123 you state: "However, resistance to anti- scabies therapy has often been recorded in the practice [50]". However, reference 50 is reviewing mechanisms of resistance and resistence is seldom evaluated in practice. I would therefor recommend to either refer to therapy failure or suspicion of resistance.

Reference 52 lacks the author

Line 132/133 is contradicting line 133/134 regarding the distribution by age.

Line 141/142 "Furthermore, recent data have confirmed a correlation between scabies incidence and population movements [4,59-62]." Does any of the referenced studies actually confirm a correlation statistically or is this just a hypothesis? Do you mean that incidence is higher in the concerned populations or that population movements lead to higher incidence in the general population? Consider rephrasing.

Line 166 should read "ages".

Line 173-177: The French study does not support the statement of increasing case numbers.

Line 233: "Furthermore, scabies transmission during the COVID-19 pandemic may have 233 been the result of increased sexual intercourse during that time, especially among younger 234 individuals who may not have respected lockdown measures [23]."This is an interesting hypothesis but  cannot be supported by a study that analyses sales data. Please add correct reference.

Author Response

Dear Reviewer,

Thank you for taking into consideration our article, we have made changes according to recommendations of reputable colleagues reviewers. Please find attached our comments:

REVIEWER 1:

However, table 1 and 2 need a thorough revision. Many information on data and results are inprecise and hard to understand or even incorrect. These are some examples:

-1. Table 1 (Central Africa): what do you mean with "an important achievement of the disadvantaged classes"?

Thank you for your comment, we have clarified the meaning (level of achievement).

-2. Table 1 (Ethiopia): what do you mean by "to fix examined factors"?

Thank you for your comment, we have explained the meaning (due to high heterogeneity across the included studies, random effects meta-analysis model was computed to fix overall prevalence and associated factors of scabies).

-3. Table 1 (Fiji): What do you mean by "clinical distribution" as a risk factor?

We have clarified the meaning ("typical clinical distribution") mentioned in their article- they thought on the findings related with scabies occurrence.

-4. Table 1 (Global): "..." in the author column. Why aren´t you reporting the more recent 2015 GBD Study?

Thank you for your comment, we have changed the reference.

-5. Table 1 (Taiwan): was "clinical severity" referring to scabies or was clinical severity of other diseases associated with scabies infections?

Patients had clinical severity status regarding scabies infestation. However, due to comments of Reviewer 2 and 3, we have changed this reference with newer publication (Liu et al., 2016).

-6. Table 2 title is referring to publications from 2000-2021 but several studies were referenced with 2022.

Thank you for your comment, we have changed to 2022.

-7. Table 2 (Augustin et al.): you report the case number increase in outpatients for 2010-2015. Then you report a much higher increase for the period 2010-2016. These numbers probably refer to inpatients but you need to state this.

That is correct, we have stated this in table, thank you.

-8. Table 2 (Reichert et al.): "patients were generally 15-24 years old" is inprecise. You probably mean that incidence was highest among 15-24 year-olds?

That is correct, we have stated this in table, thank you.

-9. Table 2 (Louka et al.): all data is referring only to health centres for refugees/asylum seekers if I am not mistaken. Your report reads like scabies infestations increased in the general populations. Please specify.

That is correct, we have stated that data report reffers to refugees/asylum seekers, thank you.

-What are the selection criteria for  the studies in table 1 ("prominent reports")? It seems to be a mixture of qualitative studies and quantitative studies on frequency, risk factors but also quality of life. I woud recommend to focus on specific topics or report more extensively (For example also studies on frequency in Brazil).

We added more data on this topic: The table presents selected prominent reports and findings on scabies occurrence in various countries published during 2000-2022.  Also, we added data that (in the manuscript) we wanted to present current data on frequency and risk factors for scabies, analyzed for a period between 2000 and 2022 (i.e. at the beginning of the 21st century), based on the available reports and data presented in Pubmed basis.

-In line 122/123 you state: "However, resistance to anti- scabies therapy has often been recorded in the practice [50]". However, reference 50 is reviewing mechanisms of resistance and resistence is seldom evaluated in practice. I would therefor recommend to either refer to therapy failure or suspicion of resistance.

Thank you for your comment, we have widen this paragraph with discussion regarding suspicion of resistance together with management of scabies and non-medical therapy (according to recommendation of Reviewer 2).

-Reference 52 lacks the author

Thank you, we have added it and collaborators.

-Line 132/133 is contradicting line 133/134 regarding the distribution by age.

Thank you, we have clarified the meaning in the first sentence (distribution by gender is equal in all ages).

-Line 141/142 "Furthermore, recent data have confirmed a correlation between scabies incidence and population movements [4,59-62]." Does any of the referenced studies actually confirm a correlation statistically or is this just a hypothesis? Do you mean that incidence is higher in the concerned populations or that population movements lead to higher incidence in the general population? Consider rephrasing.

We believe that both hypotheses are correct, therefore we have rephrased this paragraph and added new references – some of them include statistic analysis.

-Line 166 should read "ages".

It is corrected, thank you.

-Line 173-177: The French study does not support the statement of increasing case numbers.

Thank you for your comment, we have excluded the reference.

-Line 233: "Furthermore, scabies transmission during the COVID-19 pandemic may have 233 been the result of increased sexual intercourse during that time, especially among younger 234 individuals who may not have respected lockdown measures [23]."This is an interesting hypothesis but  cannot be supported by a study that analyses sales data. Please add correct reference.

Thank you for your comment, we have changed to correct reference and added more details regarding this hypothesis.

Thank you again!

Reviewer 2 Report

This is a good review. However:

- Tables 1 and 2: data from Brazil, Taiwan and Belgium are very old: please evaluate the possibility to delete them.

- Clinical features and management of scabies: this chapter is unnecessary.

- Figures 1 and 2 are unnecessary.

- Lines 117-121: add oral ivermectin (not only for crusted scabies). Please read and add the last Cochrane.

Author Response

Dear Reviewer,

Thank you for taking into consideration our article, we have made changes according to recommendations of reputable colleagues reviewers. Please find attached our comments:

REVIEWER 2:

This is a good review. However:

- Tables 1 and 2: data from Brazil, Taiwan and Belgium are very old: please evaluate the possibility to delete them.

Thank you for your comment, we have removed these references and included newly published (where they were available).

- Clinical features and management of scabies: this chapter is unnecessary.

Please see the comment of Reviewer 3: „The authors include the clinical manifestations and management of scabies in a separate section, this may be because the authors feel this is important for the readers. As a consideration regarding the management of scabies, the non-medical therapy is also important in addition to medical therapy, considering that scabies is easily contagious and spread in the community must be prevented.“

Therefore in this chapter we have included references about non-medical therapy and management of scabies at the end of paragraph.

- Figures 1 and 2 are unnecessary.

Thank you for your comment, we have deleted a part of Figure 2 (microscopic findings), however, since Reviewer 1 recommended including detailed description of pictures, we have added explanation of clinical and dermoscopic findings shown in the figure 1 and 2.

- Lines 117-121: add oral ivermectin (not only for crusted scabies). Please read and add the last Cochrane.

We have added oral ivermectin, thank you.

In addition, we checked the English.

Thank you again,

Authors

Reviewer 3 Report

Table 1:

Please pay attention to the writing on table 1. Follow to the journal guidelines about the font, font size, underline, and bold. In table 1, the writing is still uneven between one paragraph and another paragraph in terms of font, font size, underline and bold.

Table 1 and Table 2:

In line 46, the authors stated that the incidences of scabies have been reported from various countries (cited from reference no 9-20). However in table 1 and 2, the authors don’t include the reference for all the reports. Please add the reference number in table 1 and 2 according to the journal cited.

Line 69-72:

“The disease may be present with typical signs of (predominantly) nocturnal itching and lesions at certain predilection sites (the sides and webs of the fingers, wrists, axillae, areolae, umbilical area, genitalia, etc.) but may also exhibit an atypical presentation (Figure 1, Figure 2)”

1. The terms nocturnal itching can be updated to only itch according to The 2020 International Alliance for the Control of Scabies (IACS) Consensus Criteria for the Diagnosis of Scabies, look for history features part in point H1 Itch.

2. In the paragraph, it was described that Figure 2 is about atypical presentation in scabies. However, this information given doesn’t match figure 2, which is about dermoscopic and microscopic findings of scabies infestation by mite Sarcoptes scabiei.

Figure 1 dan figure 2:

Please describe the figures more detail and what findings are shown in the figure. If necessary, the arrow sign can be added to explain the specific features in the figure.

Line 75-79:

If possible, please add the references cited in every 1-2 sentences.

Line 87-89:

The genus of bacteria should be written in italic

Line 117-123:

If possible, please add more references in the paragraph

In subchapter ‘Clinical features and management of scabies’ (line 67-123)

The authors include the clinical manifestations and management of scabies in a separate section, this may be because the authors feel this is important for the readers. As a consideration regarding the management of scabies, the non-medical therapy is also important in addition to medical therapy, considering that scabies is easily contagious and spread in the community must be prevented.

Aside from clinical features and management of scabies, it is also important to know how to diagnose scabies. If possible please add the diagnosis criteria of scabies according to the updated reference, for example The 2020 International Alliance for the Control of Scabies (IACS) Consensus Criteria for the Diagnosis of Scabies.

Line 129-131:

‘The epidemiological data (based on the systematic review of population-based studies) for various 127 world regions, except North America, showed prevalence estimates ranging from 0.2% to 128 71%, with the highest prevalences in the Pacific region and Latin America [53]. Therefore, 129 geographically, scabies occurs more commonly in the developing world, tropical climates 130 and in areas with a lack of access to water [53].

In consecutive sentences in 1 paragraph, if the citation was taken from the same reference, it is better to only write the reference once or it can also be interspersed with other different references.

In subchapter ‘Trends in scabies incidence/prevalence recorded during the last two decades’ (Line 145-171)

In consecutive sentences in one or more paragraphs, if the citation was taken from the same reference, it is better to only write the reference once. If possible, please also add more references in these paragraphs.

Line 172-184:

In paragraph explaining about increasing trends in scabies incidence, it is better if the supporting data was gathered in 1 paragraph, for example, scabies trend in French, Germany, and other countries was arranged in one paragraph and if possible (line 178-184) please add more references.

Line 188-189:            

‘According to available data, the prevalence of scabies infestation is higher in refugees than in the general population [1,26]’

Please mention the specific detail where this available data was coming from, for example data in Western Europe or other countries

Line 189-192:

One German study confirmed similar findings, and, aside from the rising trend of scabies infestation reported by Augustin et al., the same trend was recognized by Reichert F et al. who reported a nine-fold increase in Germany in the period between 2009 and 2018 [21].

In this paragraph, it was stated that there were three different studies from three different authors about the trend of scabies, however the authors only add one citation from reference no 21, what about the other two studies? Please also add the other references.

Line 196 dan 198:

In consecutive sentences in 1 paragraph, if the citation was taken from the same reference, it is better to only write the reference once or it can also be interspersed with other different references.

Line 203-214:

In consecutive sentences in one or more paragraphs, if the citation was taken from the same reference, it is better to only write the reference once. If possible, please also add more references in these paragraphs.

It is better if explanation about refugee was put together in the previous paragraph and  explanation about travel was put together in one paragraph. Please check the systematical writing.

Line 242-246:

Please add more references in the paragraph.

Reference no 35-39:

Please check reference no. 35-39

If possible, please only use updated references (2013 and above) that the authors feels most suited to the statement.

In general, please check the references in the paragraph. In consecutive sentences in one paragraph, only use the citation from the same reference once. If possible, please also add more references in the paragraphs.

Author Response

Dear Reviewer,

Thank you for taking into consideration our article, we have made changes according to recommendations of reputable colleagues reviewers. Please find attached our comments:

REVIEWER 3:

-Table 1: Please pay attention to the writing on table 1. Follow to the journal guidelines about the font, font size, underline, and bold. In table 1, the writing is still uneven between one paragraph and another paragraph in terms of font, font size, underline and bold.

Thank you for your comment, we have read the journal guidelines in more details and updated both tables.

 -Table 1 and Table 2: In line 46, the authors stated that the incidences of scabies have been reported from various countries (cited from reference no 9-20). However in table 1 and 2, the authors don’t include the reference for all the reports. Please add the reference number in table 1 and 2 according to the journal cited.

We have added reference number in table 1 and 2 and we also added data that selected references are presented in the tables.

Line 69-72: “The disease may be present with typical signs of (predominantly) nocturnal itching and lesions at certain predilection sites (the sides and webs of the fingers, wrists, axillae, areolae, umbilical area, genitalia, etc.) but may also exhibit an atypical presentation (Figure 1, Figure 2)”

-1. The terms nocturnal itching can be updated to only itch according to The 2020 International Alliance for the Control of Scabies (IACS) Consensus Criteria for the Diagnosis of Scabies, look for history features part in point H1 Itch.

Thank you for your comment, we have updated to itch.

-2. In the paragraph, it was described that Figure 2 is about atypical presentation in scabies. However, this information given doesn’t match figure 2, which is about dermoscopic and microscopic findings of scabies infestation by mite Sarcoptes scabiei.

Please see comment by Reviewer 3. So, we have deleted microscopic findings (according to the comment of the Reviewer 3), therefore Figure 2 now presents only typical dermoscopic finding of scabies infestation. Also, we removed „atypical presentation“ from this sentence.

-Figure 1 and figure 2:Please describe the figures more detail and what findings are shown in the figure. If necessary, the arrow sign can be added to explain the specific features in the figure.

We have added explanation of clinical and dermoscopic findings shown in the figure 1 and 2.

-Line 75-79:If possible, please add the references cited in every 1-2 sentences.

We have added references according to the recommendation.

-Line 87-89:The genus of bacteria should be written in italic

Thank you for your comment, we have changed style to italic.

-Line 117-123:If possible, please add more references in the paragraph

We have added references according to the recommendation.

-In subchapter ‘Clinical features and management of scabies’ (line 67-123)-The authors include the clinical manifestations and management of scabies in a separate section, this may be because the authors feel this is important for the readers. As a consideration regarding the management of scabies, the non-medical therapy is also important in addition to medical therapy, considering that scabies is easily contagious and spread in the community must be prevented.

Thank you for your comment, we agree and therefore we have emphasized the need of appropriate management of scabies and non-medical therapy.

-Aside from clinical features and management of scabies, it is also important to know how to diagnose scabies. If possible please add the diagnosis criteria of scabies according to the updated reference, for example The 2020 International Alliance for the Control of Scabies (IACS) Consensus Criteria for the Diagnosis of Scabies.

Thank you for your comment, we have included this reference for explaining diagnosis criteria.

-Line 129-131:‘The epidemiological data (based on the systematic review of population-based studies) for various 127 world regions, except North America, showed prevalence estimates ranging from 0.2% to 128 71%, with the highest prevalences in the Pacific region and Latin America [53]. Therefore, 129 geographically, scabies occurs more commonly in the developing world, tropical climates 130 and in areas with a lack of access to water [53].’In consecutive sentences in 1 paragraph, if the citation was taken from the same reference, it is better to only write the reference once or it can also be interspersed with other different references.

Thank you for your comment, we have removed reference from the first sentence.

-In subchapter ‘Trends in scabies incidence/prevalence recorded during the last two decades’ (Line 145-171)In consecutive sentences in one or more paragraphs, if the citation was taken from the same reference, it is better to only write the reference once. If possible, please also add more references in these paragraphs.

We have accepted you comment and wrote reference only once. Also, we added more references in these paragraphs.

-Line 172-184:In paragraph explaining about increasing trends in scabies incidence, it is better if the supporting data was gathered in 1 paragraph, for example, scabies trend in French, Germany, and other countries was arranged in one paragraph and if possible (line 178-184) please add more references.

We have gathered everything in one paragraph.

-Line 188-189:‘According to available data, the prevalence of scabies infestation is higher in refugees than in the general population [1,26]’ Please mention the specific detail where this available data was coming from, for example data in Western Europe or other countries

Thank you for your comment, we have mentioned more specific detils in the text („According to available data, the prevalence of scabies infestation is higher in immigrants and refugees coming from Middle East and Africa than in the general Belgian and German population”).

-Line 189-192:One German study confirmed similar findings, and, aside from the rising trend of scabies infestation reported by Augustin et al., the same trend was recognized by Reichert F et al. who reported a nine-fold increase in Germany in the period between 2009 and 2018 [21].  In this paragraph, it was stated that there were three different studies from three different authors about the trend of scabies, however the authors only add one citation from reference no 21, what about the other two studies? Please also add the other references.

The sentence was maybe not clear enough, we cited 2 German studies (conducted by Augustin e al. and by Reichert et al). Thank you for your comment, we have modified the sentence to make it understandable.

-Line 196 and 198:In consecutive sentences in 1 paragraph, if the citation was taken from the same reference, it is better to only write the reference once or it can also be interspersed with other different references.

We have accepted you comment and wrote reference only once.

Line 203-214:In consecutive sentences in one or more paragraphs, if the citation was taken from the same reference, it is better to only write the reference once. If possible, please also add more references in these paragraphs.   It is better if explanation about refugee was put together in the previous paragraph and  explanation about travel was put together in one paragraph. Please check the systematical writing.

We have accepted you comment and wrote reference only once. Furthermore, we have added more references in this paragraph.

-Line 242-246:Please add more references in the paragraph.

We added additional references.

-Reference no 35-39: Please check reference no. 35-39. If possible, please only use updated references (2013 and above) that the authors feels most suited to the statement.

We have removed reference no. 36 (Heukelbach et al., 2006) and added newer references (ref. no. 38 and 39).

- In general, please check the references in the paragraph. In consecutive sentences in one paragraph, only use the citation from the same reference once. If possible, please also add more references in the paragraphs.

We have accepted you comment and wrote reference only once. Furthermore, we have added more references in some paragraphs.

Thank you again!
